Hidden shifts in allometry scaling between sound production and perception in anurans

Maria Bruna 1
Tonini João F.R. 2
Rebouças Raoni raonisreboucas@gmail.com 1 3 4
Toledo Luís Felipe 1
1 Laboratório de História Natural de Anfíbios Brasileiros, Universidade Estadual de Campinas , Campinas , São Paulo , Brasil
2 Department of Biology, University of Richmond , Richmond , VA , United States of America
3 Laboratório de Ecologia Evolutiva de Anfíbios, Universidade Federal de Juiz de Fora , Juiz de Fora , Minas Gerais , Brazil
4 Programa de Pós Graduação em Biologia Animal, Universidade Estadual de Campinas , Campinas , São Paulo , Brazil
Measey John
Electronic publication date: 2023 Nov 3
Publication date: 2023
Volume: 11
Electronic Location ID: e16322
Received 2023 May 19; Accepted 2023 Sep 29
Copyright: ©2023 Maria et al.
Copyright year: 2023
Copyright holder: Maria et al.
License: This is an open access article distributed under the terms of the Creative Commons Attribution License, which permits unrestricted use, distribution, reproduction and adaptation in any medium and for any purpose provided that it is properly attributed. For attribution, the original author(s), title, publication source (PeerJ) and either DOI or URL of the article must be cited.
License URL: https://creativecommons.org/licenses/by/4.0/

Keywords: Anura, Background noise, Morphometry, Bioacoustics, Size-relationships

Funding: São Paulo Research Foundation FAPESP #2016/25358-3; #2019/03170-0; #2020/11096-8; #2022/09659-4 National Council for Scientific and Technological Development CNPq #300896/2016-6; #140874/2019-4 Coordenação de Aperfeiçoamento de Pessoal de Nível Superior CAPES - Finance Code 001 National Science Foundation DEB-1949749 Fundação de Amparo à Pesquisa do Estado de Minas Gerais FAPEMIG #APQ-04419-22 This study was supported by São Paulo Research Foundation (FAPESP #2016/25358-3; #2019/03170-0; 2020/11096-8), the National Council for Scientific and Technological Development (CNPq #300896/2016-6; #140874/2019-4), and by the Coordenação de Aperfeiçoamento de Pessoal de Nível Superior (CAPES - Finance Code 001). João Riva Tonini received support from the National Science Foundation (DEB-1949749 to Rafael de Sá) and Raoni Rebouças received support from Fundação de Amparo à Pesquisa do Estado de Minas Gerais (FAPEMIG #APQ-04419-22) and São Paulo Research Foundation (FAPESP #2022/09659-4). The Dean’s Office of Arts and Sciences at the University of Richmond covered the article processing fees. The funders had no role in study design, data collection and analysis, decision to publish, or preparation of the manuscript.

==============================
Background

Animal communication consists of signal production and perception, which are crucial for social interactions. The main form used by anurans is auditory communication, in most cases produced as advertisement calls. Furthermore, sound perception happens mainly through an external tympanic membrane, and plays an important role in social behavior. In this study, we evaluated the influence of body and tympanic membrane sizes on call frequency across the phylogeny of anurans.

Methods

We use data on snout-vent length, tympanic membrane diameter, and dominant frequency of the advertisement call from the literature and from natural history museum collections. We mapped these traits across the anuran phylogeny and tested different models of diversification. Our final dataset includes data on body size, tympanic membrane size, and call dominant frequency of 735 anuran species.

Results

The best explanatory model includes body and tympanum size with no interaction term. Although our results show that call frequency is strongly constrained by body and tympanum size, we identify five evolutionary shifts in allometry from that ancestral constraint. We relate these evolutionary shifts to the background noise experienced by populations. Body size is important for myriad ecological interactions and tympanum size is strongly associated with female call frequency preferences. Thus, allometric escape in frog calls might arise through environmental selection such as breeding in fast flowing or soundscape competition, as well as sexual selection linked to tympanum size.

Introduction

Communication between animals involves signal production and perception. Signal production is a balance of energy cost and efficiency in information transmission, and it can be highly variable. Communication signal production can be visual (Osorio & Vorobyev, 2008), olfactory (Bossert & Wilson, 1963), tactile (Weber, 1973; Cerrone, 2019), electric (Bratton & Ayers, 1987) or acoustic (Suthers et al., 2016). In addition, signal perception is relevant for successfully accomplishing communication, which is related to the efficiency of signal transmission, and consequently influences evolution in communication systems (Endler, 1993).

Among different forms of signal production, acoustic signals are one of the most widespread across animals, present in invertebrates (Wenner, 1964) and in all classes of vertebrates (Peters & Ploog, 1973; Ladich, 2019). These signals are shaped by diverse selective pressures, such as species recognition (Claridge & De Vrijer, 1994; Gerhardt & Bee, 2007), predator pressure (Cade, 1975; Tuttle & Ryan, 1981), and sexual selection (Rebouças, Augusto-Alves & Toledo, 2020). Among the advantages of acoustic emission are the relatively fast signal transmission, orientation and its complexity. For instance, sounds can be subdivided into components such as frequency, amplitude, duration, and emission rate, which can be decoded into different information (Lopez & Narins, 1991; Morais et al., 2012). However, acoustic communication can be masked by the background noise, jeopardizing communication success (Duarte et al., 2019; Lima et al., 2022). Moreover, conspicuous acoustic signals can attract acoustically oriented predators (Tuttle & Ryan, 1981). In most cases, sound perception is closely related to receiver organs and structures, which are quite diverse. In contrast, anurans can also perceive sounds using different adaptations, such as in the tiny pumpkin toadlet from the Atlantic Rainforest, Brachycephalus rotenbergae. In this species, the inner ear (here the basilar recess) is not connected to its nervous system, suggesting that high frequency sound vibrations (as the sound of their own calls) cannot be recognized. Thus, only low frequency vibrations can be perceived, which are transmitted through bone vibrations (Goutte et al., 2017).

Anurans present a range of communication signals (e.g., Cardoso & Heyer, 1995; Toledo et al., 2015; Narins, 2019), which can be used independently or in combination depending on the behavioral context (Hartmann et al., 2005; Lourenço-de-Moraes et al., 2016; Rebouças, Augusto-Alves & Toledo, 2020; Rebouças, 2022). However, the most used signal in anurans is vocalizations (Toledo et al., 2015; Köhler et al., 2017). Anurans present several vocalization or calls types used in social contexts, such as reproductive, defensive, and aggressive calls. Advertisement calls, one form of reproductive call, are the most widespread communication strategy in anurans, which are generally emitted to attract females and guard territories (Toledo et al., 2015; Köhler et al., 2017). The variation of these calls in both spectral and temporal parameters is also diverse. Although it is well known that temporal parameters of calls, such as call rate and duration, are influenced by the environmental temperature (Lingnau & Bastos, 2007; Love & Bee, 2010; Lima et al., 2022), spectral parameters in turn are less so. Spectral parameters of anuran calls are generated by anatomical structures, and consequently constrained by the body size of the calling individual (Rebouças, Augusto-Alves & Toledo, 2020; Tonini et al., 2020). Vocalizations in anurans are produced by the contraction of trunk muscles leading the air passage from the lungs to the buccal cavity, passing through the larynx where it causes the vocal cords to vibrate and, finally, produce sounds (Colafrancesco & Gridi-Papp, 2016). These sounds are further modified by the laryngeal muscles (Gridi-Papp, 2008; Ryan & Guerra, 2014) and other related structures, such as buccal cavity and vocal sac apertures (Kime, Ryan & Wilson, 2013).

Besides call emission, call perception also plays a role in anuran social contexts. Anurans use calls to assess other individuals’ physical condition and, consequently, respond in terms of territorial defense (Foratto et al., 2021; Rebouças, 2022). Thus, the information contained in calls determines territorial segregation, reproduction, and their fitness (e.g., Giasson & Haddad, 2006; Dautel et al., 2011). For most anurans, the tympanic membrane is the first structure to capture the external sound waves, transmitting acoustic vibrations to their inner ear. In general, it is connected to the otic capsule via extrastapes and stapes, also referred to as extracolumella and columella (Van Dijk et al., 2011; Mason et al., 2015). Some studies have reported that there is a direct relationship between size and acoustic sensitivity, which means that the larger the individuals, with larger tympanic membranes, the better the sound perception (Fox, 1995; James et al., 2022). These relations are physically constrained: larger individuals also have more massive vocals chords, which tends to result in lower call frequencies (Ryan, 1988a); and larger individuals also present larger tympanic membranes, which are more prone to vibrate with sounds with lower amplitudes, which results in a more sensitive ear (Fox, 1995). Thus, an escape from these ancestral relationships must be rare, and probably a consequence of a greater selective pressure resulting from fundamental physical constraints. Also, most studies have concentrated on a few species, and a broader overview of allometric relationships between tympanum, body size, and call frequency across anurans is still lacking.

There is a general understanding that the advertisement call frequency of most anuran species is correlated with individual’s body size; i.e., the larger the frog, the lower its advertisement call frequency (Ryan, 1988a). However, this pattern was not observed for some anuran lineages (e.g., Southeast Asian ranids, Ranid frogs, Fitzinger Neotropical tree frogs and Poison frogs) that evolved to have divergent allometric relationships (Tonini et al., 2020). Moreover, some recent evidence suggests that anatomical structures closely related to communication, such as tympanic membrane, must be a constraint in the context of the relation between call frequency and sound perception (James et al., 2022). Consequently, an analysis using a phylogenetic approach to test the relation between sound emission and perception should shed light on this relationship and improve the understanding of groups that previously presented allometric escapes. This study aims to evaluate the influence of body and tympanum sizes on advertisement call frequency across the anuran phylogeny.

Materials & Methods

Bioacoustic and morphometric data

We assembled data on mean advertisement call dominant frequency, which is the call frequency with the highest energy, adult males’ snout-vent length (SVL; mm; hereafter, called simply body size) and tympanum diameter (TD; mm; hereafter, called simply tympanum size; Fig. 1) from literature and complemented with measurements from specimens deposited in the Museu de Diversidade Biológica (MDBio), Universidade Estadual de Campinas, Brazil (see Supplemental Information 1). We used dominant frequency of advertisement call because, among variables in anurans’ call, this is stereotyped and not influenced by environmental conditions, such as temperature and humidity (Köhler et al., 2017), and it is commonly used in species description (e.g., Toledo, Ribeiro & Haddad, 2007; Köhler et al., 2017; De Andrade et al., 2020). Also, we only used measurements from male individuals because they are more available in literature than measurements from females, which allowed us to perform the analysis on a larger scale. Finally, we were not able to include those species which present no visible tympanum (or even no tympanum), since measurement in this case is only possible through anatomical desiccation, which is not commonly available in the literature.

Figure 1 Diversity of tympana in Anura.

Concave tympanum of Huia cavitympanum (photo by Ulmar Grafe) (A); enlarged tympanum in Thoropa megatympanum (photo by Carlos Henrique Luz Nunes-de-Almeida) (B); regular tympanum in Hylodes cardosoi (photo by Luís Felipe Toledo) (C); tympanum whit external apparatus in Petropedetes vulpiae (photo by Václav Gvoždík) (D); reduced tympanum in Phantasmarana apuana (photo by João Luiz Gasparini) (E); and tympanum not externally visible in Cycloramphus rhyakonastes (photo by Luís Felipe Toledo) (F) (individuals present different sizes).

Many species present a sexual size dimorphism (review in Monnet & Cherry, 2002); thus, we only considered males for the analysis from type series in species descriptions. We were not able to include the information for those species for which there are only females or juveniles in the type series. In several cases, males presented oval-shaped tympana. In these cases, we considered only tympanum length for our purposes. The dataset used here is available in Supplemental Information 1 following the current nomenclature available in Frost (2023).

Phylogenetic comparative analysis

We trimmed the amphibian phylogeny (Jetz & Pyron, 2018) to include only species present in our dataset (See Supplemental Information 1). We log-transformed the data on dominant frequency, snout-vent length, and tympanum size. We estimated the phylogenetic signal of each traits using Bloomberg’s K (Blomberg, Garland Jr & Ives, 2003) and Pagel’s lambda (Pagel, 1999) in phytools (Revell, 2012). In addition, we tested the fit of three nested models using PGLS, (1) DF∼SVL, (2) DF∼SVL + TD and (3) DF∼SVL + TD + SVL*TD, and compared them using Akaike’s Information Criterion (AIC) and Bayesian Information Criterion (BIC). The best fit model was implemented in the R v. 4.0.5 (R Core Team, 2022) package bayou. The bayou package fits Bayesian reversible-jump multi-optima Ornstein–Uhlenbeck (OU) models to phylogenetic comparative data (Uyeda & Harmon, 2014). We used the bayou model to identify the location across the anuran phylogeny, support, and magnitude of shifts in intercept and slope of the scaling relationship between dominant frequency, body size, and tympanum size. Our expectation is that most frog species adhere to a background allometric scaling given the strong constraint imposed by body size on functional and anatomical traits. Here, we ask whether some frog species would represent shifts in the allometric scaling of dominant frequency with body and tympanum sizes. We used as prior a half-Cauchy distribution for a and s2, and normal distribution for b and θ. In addition, we included 0.1 of measurement error to the data. We tuned model parameters to have acceptance ratios between 0.2–0.4. We ran the models four times, each run had 10 million generations, and we used the first 30% as burn, and filtered the results to shifts with 0.75 posterior probability or higher. We check whether all runs would result in similar anuran species identified as having distinct scaling compared to most other frog species. Shifts with less than four species were not considered. Analyses and data visualization were performed in R using packages ape (Paradis & Schliep, 2019), phytools (Revell, 2012), Geiger (Pennell et al., 2014), ggtree (Yu et al., 2017).

Results

We compiled complete information (advertisement call dominant frequency, body size, tympanum size, and phylogeny) of 735 species. Body (Fig. 2A; r2 = 0.452, p < 0.001) and tympanum (Fig. 2B; r2 = 0.311, p < 0.001) sizes are inversely correlated with the dominant frequency but directly correlated with each other (Fig. 2C; r2 = 0.714, p < 0.001). Thus, large frogs tend to have large tympana and call at lower dominant frequency compared to smaller frogs, confirming the strong allometric relationship between these traits. After phylogeny is taken into account, the influence of tympanum size is attenuated in relation to the model with no phylogeny, which is shown by the difference in slope between regression lines, but it still shows significant correlation (Fig. 3). All three traits have significant phylogenetic signal for both Bloomberg’s K (KSV L = 0.27, p = 0.001; KTY M = 0.28, p = 0.001; KDF = 0.13, p = 0.001) and Pagel’s lambda (λSV L = 0.85, p < 0.001; λTY M = 0.85, p < 0.001; λDF = 0.72, p < 0.001).

Figure 2 Linear regression (solid line) and phylogenetic generalized linear squared models (dotted line) body and tympanum sizes explain 45% and 31% of the diversity of dominant frequency, respectively.

Moreover, body size explains 71% of the variation in tympanum size.

Figure 3 Measured variables on the phylogeny.

Barplot of measured variables of the 735 species included in our estimatives. Inner circle represent the Snout-vent length (SVL), mid circle represent dominant frequency and outer circle represent tympanum size. Values are log-transformed.

Our model comparison results show that models 2 (DF∼SVL + TD) and 3 (DF∼SVL + TD + SVL*TD) presented similar marginal likelihood and BIC (Table 1). Although there is a strong allometric relationship between body size and tympanum size, we consider that the simplest model with no interaction between variables provided a better fit to the data (θDf∼βSVL + βTD, Table 2), and used this model to test shifts in evolutionary allometry across anurans. Despite the great diversity of body and tympanum sizes and dominant frequency across frogs, we confirm the prior expectation that most frog species adhere to a single allometric scaling relationship. However, we identify five shifts from the evolutionary constraint imposed by body size on tympanum size and dominant frequency (Table 2, Fig. 3). In Ranidae, we observe two embedded shifts: (1) Rana and Pelophylax, and (2) Hylarana, Odorrana, Babina, and Amolops; thus, we consider, for our purposes, as a single shift shared by their most recent common ancestral. Then, in Ranidae, we observe two regime shifts: a shift comprising Huia and Meristogenys (hereafter called Asian ranids), and another shift comprising Rana and Pelophylax, Hylarana, Odorrana, Babina and Amolops (hereafter called Ranid frogs).

Table 1 Output of models.

Model selection considering only tympanum (TD, model 1), only snout-vent length (SVL, model 2), considering both (model 3) or considering both variables and the interaction between them (model 4) in relation to dominant frequency (Df). Values of p refer to specific comparison between models 2 and 3 and 3 and 4, and significant values are in bold.

Model	call	df	AIC	BIC	logLik	Test	L. Ratio	p	
1	log(Df)∼log(TD)	3	1765.00	1778.80	−879.50		–	–	
2	log(Df)∼log(SVL)	3	1603.63	1617.43	−798.82		–	–	
3	log(Df)∼log(SVL) + log(TD)	4	1583.61	1602.01	−787.81	2 vs 3	22.02	<0.001	
4	log(Df)∼log(SVL) + log(TD)
+log(SVL) * log(TD)	5	1577.24	1600.24	−783.62	3 vs 4	8.37	0.004	

Table 2 Regime shifts.

Model estimates of slope and intercept for the evolutionary regime shifts in dominant frequency (DF), tympanum (TD), and body size (SVL).

Taxa	θ DF	β TD	β SV L	Posterior
probability	
Root	9.338	−0.369	−0.4034		
Fitzinger Neotropical treefrogs	8.507	−0.179	−0.0412	0.81	
Neotropical swamp frogs	7.821	−0.185	−0.1371	0.96	
Neotropical torrent frogs	8.078	0.025	0.0999	0.95	
Asian ranids	8.543	0.048	0.1383	0.86	
Ranid frogs					
Rana	7.235	0.001	−0.1159	0.76	
Amolops, Babina, Hylarana, Pelophylax, and Odorrana	8.402	−0.073	−0.0828	0.68	

Among other regime shifts, in Dendropsophus (hereafter Fitzinger Neotropical Treefrogs) and Leiuperinae (hereafter Neotropical swamp frogs) we observe a negative slope, as most of anurans, but with a different intercept for the allometric relationship. In Fitzinger Neotropical treefrogs, sound frequency is decoupled from body size but still negatively correlated to tympanum size, while in Neotropical swamp frogs the dominant frequency is associated with body and tympanum size (Table 2). In Hylodidae (hereafter Neotropical torrent frogs), Asian ranids, and Ranid frogs, the evolution of dominant frequency is decoupled from the constraint of body and tympanum size, which is shown by the zero slope; whereas in Ranid frogs we observe the inverse situation, in which sound frequency is decoupled from tympanum size but still persists dependent on body size. In Asian ranids, sound frequency is dissociated from tympanum size and positively correlated to body size, which is unique and indicates that large species within those genera tend to call at higher frequency as opposed to what is expected for most other frog species (Figs. 4 and 5).

Figure 4 Allometric shifts across anuran phylogeny.

Relation between dominant frequency and body size (left chart) and between advertisement call dominant frequency and tympanum size (right chart). The general relationship for all sampled species is in grey, and specific relation are coloured as follows: Asian ranids (black), Ranid frogs (red), Neotropical torrent frogs (blue), Fitzinger Neotropical treefrogs (green) and Neotropical swamp frogs (orange). Correlations are not significant.

Figure 5 Estimates of intercept and slopes of model.

Density plots showing the uncertainty in model parameter estimates of intercept (θDominant frequency) and slopes (βSnout-vent length and βTympanum size) for each escaped lineage: Hylodidae (blue), Dendropsophus (green), Leiuperinae (orange), other ranids (red) and Huia + Meristogenys (black).

Discussion

Frogs have a wide range of body and tympanum size and call frequency, as well as a great diversity of reproductive behaviors (Haddad & Prado, 2005; Nunes-de-Almeida, Haddad & Toledo, 2021). In addition, frogs have colonized a variety of environments across all continents except Antarctica. Despite the vast environmental complexity in terms of biotic and abiotic factors presenting a myriad of selection pressures, the constraint of body size overcomes those selective pressures and strongly constrain the relationship between sound frequency and tympanum size. Our results show that most frog species adhere to a single allometric scaling relationship between advertisement call dominant frequency, body, and tympanum size. Although previously reported for the relationship between call dominant frequency and body size (Tonini et al., 2020; James et al., 2022), we observe here allometric escapes for the relationship between dominant frequency and tympanum size as well (e.g., Rebouças, Augusto-Alves & Toledo, 2020).

In at least four groups, shifts appear to be independent, considering the great phylogenetic distance between them (Fig. 3)—out of the five shifts observed here, two of them are within ranids. Additionally, our results include allometric shifts for three of four previously reported groups (Asian ranids, Fitzinger Neotropical treefrogs, and ranid frogs) (Tonini et al., 2020). We did not observe any allometric shift including tympanum size for Neotropical poison frogs, as previously reported, and we estimate a shift for Neotropical swamp frogs and Neotropical torrent frogs, which were not observed previously (Tonini et al., 2020). Thus, considering that Tonini et al. (2020) only evaluated variables related to sound emission and James et al. (2022) evaluated the tympanum size allometry (linked to sound perception) of a reduced number of species (81 spp., with little overlap for all measurements), our study represents a broader estimation of allometric shifts across the anuran tree of life.

Our results show that call frequency is dependent on the size of individuals and correlated with tympanum size. We agree that the relations observed here are referred to interspecific relation, but as we observed shifts for groups including several species, we hypothesize that these effects could be a result of interaction between individuals with the environment along the time, which consequently is reflected in the relation between species. Advertisement calls in anurans, mostly emitted by males, are used both to attract mates and to segregate calling males (Toledo et al., 2015). Accordingly, we suggest two sets of limiting factors: internal, which constrain the vocalization emission, such as body size and other physiological implications (Köhler et al., 2017); and external, which constrain the understanding of social context through the calls of other males, such as inner ear structures, amphibian papilla (for lower frequencies), and basilar papilla (for higher frequencies) (Schoffelen, Segenhout & Van Dijk, 2008). Among species groups representing allometric shifts, Fitzinger Neotropical treefrogs and Neotropical swamp frogs showed a similar negative relation between tympanum size and dominant frequency (i.e., for sound sensitivity). This relationship was distinct from all other groups (Asian ranids, Ranid frogs and Neotropical torrent frogs), which possibly indicates different selective pressures for sound perception. In relation to sound emission (i.e., size and dominant frequency), for Fitzinger Neotropical treefrogs, the sound frequency was decoupled from body size and Neotropical swamp frogs remained size dependent with species calling at lower frequency than expected. For Ranid frogs (referred as Rana and Pelophylax, Hylarana, Odorrana, Babina, and Amolops in Table 1), it is similar to Fitzinger Neotropical treefrogs and Neotropical swamp frogs, but for Neotropical torrent frogs and Asian ranids, the relation was inverse, with larger individuals presenting higher dominant frequencies.

Some causes of allometric shifts might be common for all groups of frogs in a certain way. For example, species of Asian ranids, ranid frogs, and Neotropical torrent frogs call near waterfalls and fast flowing water bodies. These environments are highly noisy, and consequently, over the time can limit calls to frequencies higher than the background noise (Tonini et al., 2020). In Crossodactylus schimidti, for example, males show a short-term adjustment for dominant frequency in face of background noise frequency (Vidigal et al., 2018). Similar results were found for other species in the genus (e.g., C. gaudichaudii and C. werneri) and for most of Hylodes species as well (e.g., H. charadranaetes, H. glaber, and H. malhagaesi) (Augusto-Alves, Dena & Toledo, 2021). Among species of the other two groups recovered as a shifts, Huia cavitympanum for example present most of their communication through ultrasonic calls, ranging from 5 to 25 kHz (Arch, Grafe & Narins, 2008), as do Wijayarana masonii (formerly H. masonii) (Boonman & Kurniati, 2011). Similar results were also found for Amolops tormotus (Feng et al., 2006) and Odorrana graminea (Shen et al., 2011).

The Fitzinger Neotropical treefrogs and Neotropical swamp frogs are not known to call in fast flowing water environments, but our results also indicate them as an allometric scape from the ancestral body size constraint. Species of these two groups use ponds and swamps as reproductive sites. Species in both groups reproduce year around, frequently with hundreds of individuals calling at the same time very closely from each other in lek (Barreto & Andrade, 1995; Camargo et al., 2005; Abrunhosa, Wogel & Pombal, 2006; Curi et al., 2014; Pompeu, de Sá & Haddad, 2020). In this case, we hypothesize that the selective force is not fast flowing water but the noise produced by other organisms, including vertebrates and invertebrates. In this scenario, this resulting soundscape may impose a similar selective pressure to individuals as fast flowing streams (e.g., Both & Grant, 2012; Lima et al., 2022). The environment in which these species tend to live might also influence the optimal call frequency (Marten, Quine & Marler, 1977; Fricke, 1984). For a given body size, individuals of Fitzinger Neotropical treefrogs and Neotropical swamps frogs call at a lower frequency than expected, which might represent an advantage in territorial dispute and female attraction. The fitness of males is commonly evaluated through call for both males and females (Rebouças, Augusto-Alves & Toledo, 2020; Rebouças, 2022), since lower-frequency calls indicate larger males (Tonini et al., 2020), which may present an advantage in disputes (Ryan, 1988b). Consequently, larger males, which spend less energy calling in lower frequencies, present a higher probability to win disputes, and consequently better protect the reproductive territory. In this scenario, females tend to be more attracted by those males through lower-frequency calls (Ryan, 1988b). In some cases, males could even present a call with a frequency lower than predicted by its size, which effort the sexual selection role of calls (Rebouças, Augusto-Alves & Toledo, 2020). In these two groups, the selective constraint of body size on sound frequency and tympanum size is weaker compared to other frogs, which is shown by the lower slope value relative to the background regime. Once sound frequency and tympanum size are less constrained by body size, call frequency does not necessarily indicate larger sizes. Therefore, selection on tympanum size and male sound frequency could result from other parts of the male advertisement call, such as temporal parameters or behavioral interactions that include territorial or mating displays. However, an intraspecific relationship between call frequency/body size/tympanum size could still be present, and further analysis should evaluate within species variation, mainly in these groups that presented allometric escapes.

Conclusions

This study evaluated allometric escape across the anuran phylogeny using parameters of call emission (i.e., males’ call frequency) and parameters of call sensitivity (i.e., tympanum size). We showed that the inclusion of tympanum size allows the identification of new acoustic allometric shifts across anurans. Also, we hypothesize that shifts might result from selective pressure of background noise and those that reproduce in high species-rich or overpopulated ponds. Finally, our observations provide insights for future studies which aim to evaluate sound communication in anurans, and additional conclusions could be reached with measurements of females in the analysis.

Supplemental Information

Supplemental Information 1 Morphometric and bioacoustics dataset for inference of allometric shifts

Click here for additional data file.

We thank to Victor Fávaro, Isabel Máximo, and João Pedro Bovolon Thomaz for their help in accessing individuals in museum; Ulmar Grafe, Carlos Henrique Luz Nunes-de-Almeida, Václav Gvoždík, and João Luiz Gasparini for kindly allowing us to use their photos, and Alexander Pyron for the review of the manuscript. We also thank the Museu de Diversidade Biológica for allowing us to access the specimens under their care.

Additional Information and Declarations

Competing Interests

Author Contributions

Data Availability

The authors declare there are no competing interests.

Bruna Maria conceived and designed the experiments, performed the experiments, authored or reviewed drafts of the article, and approved the final draft.

João F.R. Tonini analyzed the data, prepared figures and/or tables, authored or reviewed drafts of the article, and approved the final draft.

Raoni Rebouças conceived and designed the experiments, performed the experiments, analyzed the data, prepared figures and/or tables, authored or reviewed drafts of the article, and approved the final draft.

Luís Felipe Toledo conceived and designed the experiments, authored or reviewed drafts of the article, and approved the final draft.

The following information was supplied regarding data availability:

The raw data is available in the Supplementary File.

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
