# Peer review of "Hidden shifts in allometry scaling between sound production and perception in anurans"

_PeerJ, doi:10.7717/peerj.16322_

## Round 0.1 · original submission · Major Revisions

Your manuscript has been reviewed by three reviewers all of whom see merit in your impressive dataset and subsequent analyses. However each of them raised points concerning clarity of the analyses and the way in which it is reported that you need to respond to in detail in your revised ms.

In particular, Revs #1&3 note that evolutionary consequences of the allometric shifts reported in the study are still unclear. I also found the overall conclusion of the study somewhat obscure. Rev #3 also notes that your phylogenetic coverage of the analysis is not very explicit, and needs to be clarified.

I look forward to a revised ms that carefully considers these three excellent reviews.

Reviewer 1 ·

Basic reporting

Some fixes to the language throughout the manuscript are needed. I have made detailed suggestions for the abstract, and similar logic could be applied to the other sections.

Background:
Sentence 1: consists “of”, not “in”
Sentence 1: within animal communication, the terms “signal production and perception” are more intuitive to me, but there may be a good reason the authors prefer “emission and reception”
Sentence 2: it is not clear what context the readers are referring to with “in this context”
Sentence 3: “However” as the opening word does not actually juxtapose anything from the previous sentence. The authors have already made it clear that sound perception is important from the first sentence (although, I would suggest harmonizing the use of terms here, rather than using reception and then perception).

Methods:
Sentence 2: “and tested”

Results:
I do not understand the sentence construction of: “sound emitted by other conspecific or not, but syntopic individuals”
“Thus” at the beginning of the last sentence does not make sense to me.
“might arise” or “might have arisen”


There are also quite a few cases of verb/tense agreement and other fixes that I would recommend.
I have a few suggestions here, and I would be able to provide additional suggestions in a future draft.
L144: should be “dominant frequency of the advertisement call” rather than “advertisement call of dominant frequency”

L145: “as a character to evidence a difference between species” does not make sense to me. This could just be deleted.

L169: I do not understand how the shift is “hidden”

L186: “the strength of tympanum size is attenuated” – in relation to what?

A bit confused about using the term “recover” when referring to observing an allometric shift.

L301: rheophilic refers to a property of the animal (liking rivers) rather than the environment. Regardless, this term is not very commonly used, so it should either be defined at some point, or replaced.

Methods reference supplementary material – the only material I had was the dataset, which does not make sense for some of the references.

Experimental design

The authors have amassed an excellent dataset. I would appreciate a few clarifications for the reader about the analyses:

The authors conclude that allometric escapes are rare. This is something I have often been confused about – the analyses look for groups of animals that have a relationship that is different from the typical relationship. Doesn’t this mean that all allometric escapes must be rare? Because it is inherently analyzing for deviations from the “typical” relationship?

While I am not very familiar with these exact statistical methods, I am quite confused by the data plotted in Fig 4. The trendlines do not seem to match the data very well for almost any of the groups. I realize that, with the phylogenetic models, data points are not treated independently so the trendlines may not look quite like a linear model. However, I cannot figure out how a trendline could fail to even intersect any of the data (as seen in the black dataset on the right). Furthermore, the actual data in green looks to follow the allometric scaling of the grey background data quite well, so I am surprised by how flat the trendlines ae. Perhaps additional explanation of these analyses targeted at non-experts would be useful.

Validity of the findings

In the Results section of the abstract, the authors write about different types of background noise that could help explain their results. However, this is not actually in the results section of the manuscript, and is, rather, simply posited in the discussion. If the authors wish to propose this, I believe they need to justify this with at least some attention in the results. For instance, if all species are categorized as leking or as calling in fast flowing streams, the authors can run analyses to see if these actually do predict allometric escape. I do not expect this to pan out, and a lot of species of frogs have to deal with all kinds of background noise. Without justification in the results, I think these ideas need to be de-emphasized, given that there are myriad possible explanations (in addition to random chance).

It is also unclear to me how “allometric escape would allow an advantage in territorial dispute and female attraction”. Territorial disputes and female attraction tend to occur within species, whereas all of the data here are presented across species. I think this needs to be explained further, and presented as a possibility, rather than a statement of fact. Similarly, the authors also state "Once sound frequency and tympanum size are less constrained by body size, call frequency not necessarily would indicate larger sizes, and species could use color or behavior displays to also attract females" – but females would be assessing this relationship within her species, not across species.

Conclusions are really about signal reception, but I think they need to be tempered by the fact that tympanum size of females was not considered.

Reviewer 2 ·

Basic reporting

The ms is on relationship between body size, tympanic size and call frequency in a number of frog species. Using statistical analysis across species in frog lineages it is shown that some groups deviate from the most obvious (allometric) relationship.

The text needs some revision for clarity (as indicated in the attached pdf).
I found the taxonomy unclear, also in the supplementary excel file.
Figures and tables OK

I think the authors could do a better job delineating the basic hypotheses. The size/frequency relationship has a solid physical basis - larger structures have larger mass that will tend to lower the emitted frequency.
Also, larger tympana are generally more sensitive, especially at lower frequencies. These are physical facts that the empirical findings can be held up against.

Experimental design

The ms is within aims and scope.

I am not sure whether the research is relevant. It appears to me that this type of correlation does not take large intraspecies variation within the families into account.

Investigations OK

There are details of the methods I think could be reported clearer: how do the authors deal with species where there are no clearly visible tympanum (like in figure 1F) or in genuinely 'earless' species? Also, the earless Bombina is erroneously listed as a ranid frog - I am not sure how that affects the results.
I cannot evaluate the R-models, but assume this part is OK.

Validity of the findings

I have problems about the validity of the findings. I think this large scale analysis obscure the large variations within a group for example of Asian ranids. Some of these frogs (Odorrana) have ultrasonic calls and their middle ear and vocal apparatus is greatly modified compared to the more 'ordinary' ranids that have quite different habitats. I think averaging the properties across these very different adaptations (and probably different breeding strategies as well) is not likely to be really informative.

I could not understand the taxonomy in the supplementary excel sheet. It looks like almost every species is listed in the superfamily hyloidea, whereas ranids, microhylids and rhacophorids should be in ranoidea, pelodytes in pelobatoidea etc. I do not know whether this unclear taxonomy has any influence on the model results.

Additional comments

Please see more specific comments in the attached pdf.

Annotated reviews are not available for download in order to protect the identity of reviewers who chose to remain anonymous.

Reviewer 3 ·

Basic reporting

The English used is of good quality. The cited literature is consistent with the published literature. The structure of the article is correct. The database used for the analyzes is shared.

Experimental design

Research question is well defined, relevant and meaningful. Overall, the methods are well described but some information is still missing (see attached document).

Validity of the findings

Please see the attached document.

Annotated reviews are not available for download in order to protect the identity of reviewers who chose to remain anonymous.

---

## Round 0.2 · Minor Revisions

Thanks for your resubmission. One reviewer has now looked at this ms and although we are broadly happy with the way that you have revised your submission, there remain some few outstanding points that require clarification. In addition to the grammatical errors pointed out, please could you respond to the following:

Abstract: Remove sentence: "We recognize two sources of background noise causing allometric shifts: i) environmental noise due to fast flowing water bodies and ii) biotic noise due to the sound emitted by other syntopic individuals." You can state that you propose hypthoses that can be tested - but I agree that to detail your speculative ideas in the abstract is beyond credibility.

L230 & L303: Allometric escape - I suggest making this into a discussion point that includes the competing views (intra- vs inter- specific). Make it clear how you feel your data supports one over the other.

Figure 4: I agree with the reviewer that the correlation lines plotted for the groups shown in detail appear not to match the data. Note that you should only plot lines for significant relationships (- I doubt that Asian ranids could have a significant relationship with such a small data set). If no significant relationship exists do not plot a line. Note that Neotropical swamp frogs appear to have 2 different colours.

Reviewer 1 ·

Basic reporting

Much of the wording has been improved, however, there continue to be quite a few grammatical errors. I have tried to catch as many as I can, but I am sure I missed some.

l. 133: should be "call" not "calls"
l. 134: should be "highest" not "higher"
l. 134-135: don't need to say "male" twice
l. 139: should be "calls" not "call"
l. 143: should be "on a larger" rather than "in large"
l. 145: should be "is" not "are"
l. 182: the 2 in r2 should be superscript
l. 298: should be "call frequency does not necessarily indicate"
l. 303: remove "it considering an"

l. 280: this needs to be rephrased: "In this case, the selective force causing is not fast flowing water but hypothesized to be the noise of other organisms that produce sounds, including vertebrates
In addition, should say "we hypothesize" to make it clear that this is not based on any actual result

l. 73: "can improve signal sensitivity in such a context."
It is not clear what context the authors are referring to.

l. 77: "However, low frequency vibrations can be perceived through their arms, which transmit sound to the inner ear through bone vibrations (Goutte et al., 2017)."
I don't believe this reference discusses such a novel perceptual mechanism.

Experimental design

no comment

Validity of the findings

A few of my questions/concerns regarding the conclusions I think could use just a bit more attention:

The response to my question about figure 4 did not clarify for me what the results are depicting.
The legend says: "The general relationship for all sampled species is in grey, and specific relation are coloured as follows"
However, the colored lines often do not match the data well at all.

I think the abstract should not reference the hypotheses for the sources of the allometric shifts because these are purely speculative with no data to support.
In particular, the statement: "We recognize two sources of background noise causing allometric shifts" is not justified by the data in the manuscript.

I still remain unconvinced by the argument that allometric escapes can be explained by sexual selection for lower frequency calls. That explanation only works on within species comparisons, because members of different species will not be competing with each other to have lower calls than expected for their body size (which is what allometric escape comparisons look at).

---

## Round 0.3 · accepted · Accept

I'm happy to accept this manuscript once the following changes are made:

1. Remove the 2 hypotheses from the abstract - as you agreed to do. (it's fine to hypothesise in the discussion)

2. L230 is unchanged so there still needs reference to the intraspecific discussion point. For example, you can add "(but see below)" at L203.